

# Influence of topography and human activity on erosion in Yunnan, SW China

Amanda H. Schmidt[1], Thomas B. Neilson[2], Paul R. Bierman[2,3], Dylan H. Rood[4,5,6], William B. Ouimet[7], Veronica Sosa Gonzalez[3]

[1] Geology Department, Oberlin College, 403 Carnegie Building, 52 W. Lorain St., Oberlin, OH 44074, USA

[2] Department of Geology, University of Vermont, 180 Colchester Ave., Burlington, VT 05405

[3] Rubenstein School of Environment and Natural Resources, University of Vermont, Burlington, VT 05405

[4] Department of Earth Science and Engineering, Imperial College London, South Kensington Campus, London SW7 2AZ, UK

[5] AMS Laboratory, Scottish Universities Environmental Research Centre, East Kilbride G75 0QF, UK

[6] Earth Research Institute, University of California, Santa Barbara, CA 93106, USA

[7] Department of Geography and Center for Integrative Geosciences, University of Connecticut, Storrs, CT, 06269, USA

*Correspondence to*: Amanda H. Schmidt (aschmidt@oberlin.edu)

**Abstract.**

In order to understand better if and where long-term erosion rates calculated using *in situ* [10]Be are affected by contemporary changes in land use and attendant deep regolith erosion, we calculated erosion rates using measurements of *in situ* [10]Be in quartz from 52 samples of river sediment collected from three tributaries of the Mekong River (median basin area = 46.5 km$^2$). Erosion rates range from 12 – 209 mm/kyr with an area-weighted mean of $117 \pm 49$ mm/kyr (1 standard deviation) and median of 74 mm/kyr. We observed a decrease in the relative influence of human activity from our steepest and least altered

watershed in the north to the most heavily altered landscapes in the south. In the areas of the landscape least disturbed by humans, erosion rates correlate best with measures of topographic steepness. In the most heavily altered landscapes, measures of modern land use correlate with [10]Be-estimated erosions rates but topographic steepness parameters cease to correlate with erosion rates. We conclude that in some small watersheds we sampled, those with high rates and intensity of agricultural land use, that tillage and resultant erosion has excavated deeply enough into the regolith to deliver subsurface

sediment to streams and thus raise apparent *in situ* [10]Be-derived erosion rates by as much as 2.5 times over background rates had the watersheds not been disturbed.

## 1 Introduction

Understanding the source and volume of sediment moving across the landscape, and the role of humans in sediment generation and transport, are fundamental issues in Earth Science (NRC, 2012). *In situ* [10]Be ([10]Be$_i$; $t_{1/2}$ = 1.39 My) (Chmeleff

et al., 2009), measured in detrital quartz sand, is widely used (Portenga and Bierman, 2011) to estimate basin-wide erosion rates integrated over $10^3 – 10^5$ years (Brown et al., 1995;Bierman and Steig, 1996;Granger et al., 1996). The technique is useful even in areas disturbed by people (1995;Brown et al., 1998;Granger et al., 1996;Bierman and Steig, 1996;Hewawasam



et al., 2003;Reusser et al., 2015;Vanacker et al., 2007), because human activity does not typically erode sediment below the mixed layer over the scale of sampled watersheds.

Despite the fact that humans are now one of the most effective geomorphic agents (Hooke, 1999, 2000), most prior research measuring $^{10}Be_i$ in river sediment to document erosion rates has relied on the assumption that human disturbance in

most environments does not alter calculated erosion rates. However, sediment gauging records are sparse and problematic for constraining the effects of human activity on erosion rates (e.g., Trimble, 1977) because of both the time frame of integration and because of in-catchment storage, both hillslope and alluvial. Land use changes are often not of large enough scale nor do they usually occur during discrete enough time periods so as to affect an entire upstream watershed; thus, it is typically not possible to untangle the relative effects of human activity and natural factors, such as topography and climate,

on erosion rates calculated using the measured abundance of $^{10}Be_i$ in fluvial sediment. Understanding when and where human activity has changed the concentration of $^{10}Be_i$ in contemporary river sediment and thus increased cosmogenically-based erosion rates will suggest where such rates are likely to be most accurate and least biased.

Southwestern China is an ideal location to test the effects of focused change in upstream land use over short periods of time due to top-down land use policies that were widely implemented throughout the region over the last 50 years. From

the 1950s to the late 1980s, western China experienced what is termed as the Three Great Cuttings (the Great Leap Forward [late 1950s], Grain as the Key Link [early 1970s], and Opening and Development [mid-1980s]) (Shapiro, 2001;Trac et al., 2007), which are blamed for widespread erosion on very small (plot) scales (Urgenson et al., 2010;Zhang, 1999;Zhang and Wen, 2002, 2004). Thus, if anywhere is likely to show an effect of modern increases in agriculture on apparent long-term erosion rates, it will be southwestern China.

In this paper, we report long term $^{10}Be_i$-derived erosion rates for a series of nested samples contained within three watersheds in Yunnan Province (Figure 1). By focusing in detail on small subcatchments within three small (200-2500 km$^2$) watersheds, we are able to understand better how differences in precipitation, topography, and land use influence long term $^{10}Be_i$ erosion rates in tropical regions in southwestern China.

## 2 Field Sites

We chose the Yongchun, Weiyuan, and Nankai Rivers based on the range in basin area (200 – 2500 km$^2$), relative position in the regional N-S gradient in rainfall (Figure 1C) (Fan et al., 2013), and high topographic variability (Figure 1B).

The Yongchun River watershed, situated on the southeastern margin of the Tibetan plateau (Figure 2), is a small (198 km$^2$) high-elevation watershed, with the lowest mean annual precipitation (MAP = 869 mm/yr) of the study sites (Figure 1c). Based on field observations of river-borne clasts and the available geologic map, the basin is underlain by

Triassic granite, shale, sandstone, and limestone, and Neogene sandstone, mudstone and conglomerate; significant portions of the basin are mantled by Quaternary alluvium (Ministry of Geology and Mineral Resources, 1986), but the geologic map is not diagnostic of quartz content, making it impossible to determine the relative distribution of quartz in the watershed. The





Yongchun River bifurcates into northern and southern arms, with the steeper sub-basins in the southern arm and high-elevation, low-slope surfaces in the northern arm (Figure 3). In 2012, a large (~30 m tall) dam was completed in the southern arm of the Yongchun and we observed numerous small diversion and check dams as well as out-of- and in-channel gravel mining operations. We sampled upstream of the large dam to minimize effects on our samples. Land use in the Yongchun basin consists primarily of forest, cultivated land, shrubland, and grassland.

The topography of the Yongchun is unique among the basins we sampled, with incision along the main-stem of the Mekong River overprinting the influence of normal faulting in the basin. The Yongchun watershed was part of a regional high-elevation, low-relief surface formed during the Oligocene to early Pliocene (Clark et al., 2006;Liu-Zeng et al., 2008). A mapped NW-SE trending fault offsets formations mapped as Neogene (Ministry of Geology and Mineral Resources, 1986), with normal motion, inferred from field observations, occurring after the formation of the low-relief surface. The main- and southern-arms of the Yongchun River currently flow along the fault trace (Figure 3). Faulting increased relief in the southwestern footwall portion of the Yongchun basin, and fault scarp knickpoints migrated up the southwestern drainages forming the steep, high-elevation topography currently observed. Headward migrating knickpoints from the Mekong River entered the Yongchun River after ~9 – 13 Ma (Clark et al., 2005) and began the ongoing process of eroding the remaining low-relief landscape on the hanging wall. Knickpoint propagation was likely faster along the main- and southern-arms of the river, where incision was facilitated by the fault, resulting in less low-relief area on the hanging wall in the southern arm.

The Weiyuan watershed is the largest of the three basins (2508 km$^2$), further south in the regional rainfall gradient (MAP = 1050 mm/yr), and lower in elevation than the Yongchun basin (Figure 1). The Weiyuan watershed consists of a main-stem river that is joined by the western arm near the outlet and the eastern arm ~20 km upstream of the outlet (Figure 2). Steep slopes generally prevail throughout the basin with gentle slopes limited to valley floors (Figure 3). Field observations and geologic mapping indicate that the majority of the basin is underlain by Paleogene mudstone, sandstone, and conglomerate, with Cretaceous sandstone and siltstone in the western arm and northern-most portion of the basin (Ministry of Geology and Mineral Resources, 1986); as with the Yongchun, we are unable to determine the relative abundance of quartz throughout the watershed from the geologic map. The western arm of the Weiyuan River holds the largest of the dams in the studied basins (>30 m tall), which was completed in 1990. The dam is >35 km upstream of the nearest sample on the western arm of the Weiyuan River, limiting the influence of this dam. We also observed many mid-sized and smaller dams and diversions throughout the watershed, as well as numerous active-channel gravel mining operations. Land in the Weiyuan watershed is generally either forested or cultivated, with agriculture comprising ~ 22% of the total basin area, primarily in valley bottoms (Figure 3).

The Nankai River watershed (1006 km$^2$) is the lowest elevation, furthest south, and wettest of the three basins (MAP = 1299 mm/yr; Figure 1b). Late Paleozoic-Mesozoic quartz monzonite and Proterozoic low- to mid-grade metamorphic rocks primarily underlie the basin; however, the northernmost portion of the basin is underlain by Jurassic sandstone and siltstone, and valley bottoms are covered by Quaternary fill (Ministry of Geology and Mineral Resources, 1986). Steeper upland sub-basins and expansive low slope valley floors characterize the Nankai watershed (Figure 3). The





majority of streams are diverted near the mountain-front to irrigate sugarcane and rice paddies that cover the valley floor; the natural river channel in much of the northern arm is completely obscured by agriculture and irrigation structures (Figure 2). At least four dams have been constructed in the upland sub-basins of the watershed, but only affect very small subbasins. Mining of active channel gravel was common where the natural river channel was present. The landscape is primarily

cultivated, with some forest (primarily rubber plantations) and grass/shrubland (primarily tea plantations) (Figure 3).

# 3 Methods

## 3.1 Sampling

Using GIS and remotely sensed data, we selected 52 in-channel sample sites in three different drainage basins (Figure 4, Table S1). Sampled upland sub-basins (n = 25) include the full range in mean slopes across all sub-basins over ~5

$km^2$ and include basins with end-member land uses (i.e., primarily cultivated or forested). We also collected a series of samples along major trunk-streams between the uplands and outlet in each basin (n = 25).

We collected samples of fluvial sediment from point bars, mid-channel islands, depositional pools, and channel beds in 2013 immediately prior to the start of the summer monsoon. Sample sites were re-evaluated in the field to account for intensive human alteration of the channel nearby, including sediment mining. If human alteration was present, we moved

sampling sites to a more suitable location (usually upstream). We field sieved sediment to 250 – 850 μm.

## 3.2 *In situ* [10]Be extraction and measurement

Bulk aliquots of each sample were purified to isolate quartz using chemical etching (Kohl and Nishiizumi, 1992) at the University of Vermont. Prior to Be extraction, the purity of isolated quartz was tested using inductively coupled plasma – optical emission spectroscopy. Be was extracted from ~5 – 25 g of purified quartz spiked with ~250 μg of beryl carrier

following established procedures (Corbett et al., 2016). Each batch included one process blank and one CRONUS N standard (Jull et al., 2015). $^{10}Be/^9Be$ ratios were measured by Accelerator Mass Spectrometry at the Scottish Universities Environmental Research Centre (Xu et al., 2010;2015), normalized to the NIST standard with an assumed $^{10}Be/^9Be$ ratio of 2.79 x $10^{-11}$ (Nishiizumi et al., 2007), and background corrected by the average process blank ratio of 2.64 ± 0.98 x $10^{-15}$ (n = 7, 1 SD) (Table S2). The single replicated field sample agreed to < 1%. Erosion rates were calculated based on the $^{10}Be_i$

abundance in each sample, effective elevation (Portenga and Bierman, 2011), mean latitude, and mean longitude using the CRONUS-Earth online calculator (Accessed March 2014; main code v2.2, constants file v2.2.1, global production rate, and the time invariant Lal/Stone scaling model Table S3) (Balco et al., 2008).



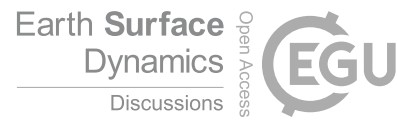

### 3.3 Digital data

We use 30 m resolution digital elevation models generated by NASA and METI's ASTER GDEM program (NASA LP-DAAC, 2012a) as the basis for calculating total basin relief, slope, and normalized channel steepness ($k_{sn}$). Relief is the difference between the minimum and maximum elevation for each basin. We use precipitation data provided by the APHRODITE program, a collaboration between the Research Institute for Humanity and Nature Japan and the Meteorological Research Institute of Japan Meteorological Agency. We use version APHRO_MA_V1101, which consists of daily 0.25° gridded precipitation data from 1951 to 2007 (Yatagai et al., 2012). Although it does not have the highest spatial resolution, APHRODITE provides the most accurate rainfall estimates of available datasets for this region (Andermann et al., 2011). Land-use data is from the GLC30 land cover dataset and represents 30 m resolution land cover from 2010 derived from Landsat TM, ETM+, and Chinese HJ-1 multispectral satellite images and a suite of auxiliary data sources (Chen et al., 2015). Our study watersheds are dominated by agricultural, forested, shrubland, and grassland (only Yongchun has grassland). We combine shrub/grasslands and use these three categories as our primary metrics of land use.

We calculate channel steepness using longitudinal river channel profiles derived from the DEM. The channel slope (S) and drainage area (A) of a fluvial channel are typically related through the power-law (Flint, 1974) (eq. 1):

$$S = k_s A^{-q} \qquad (1)$$

where $k_s$ is the steepness index and θ is concavity. Channel steepness index, $k_s$, is highly sensitive to variation in θ, a complication we correct for by using a reference concavity for all basins of θ = 0.45, allowing us to derive normalized channel steepness ($k_{sn}$) in place of $k_s$ (Wobus et al., 2006). We average $k_{sn}$ in all channel segments (1 km long) where A > 1 km$^2$, and present basin-wide mean and median $k_{sn}$, as employed by other studies (DiBiase et al., 2010;Miller et al., 2013;Ouimet et al., 2009). We use median $k_{sn}$ in analyses to minimize the effects of outlier values but present both in supporting material.

### 4 Results

Across all three study basins, $^{10}$Be$_i$-derived erosion rates range from 12 – 209 mm/kyr with an area-weighted mean of 117 ± 49 mm/kyr and median of 74 mm/kyr (Figure 5). In the Yongchun River watershed, erosion rates vary from 12 – 209 mm/kyr, with an area-weighted mean of 51 ± 57 mm/kyr and median of 38 mm/kyr. Erosion rates in the Weiyuan watershed are generally higher, from 55 – 193 mm/kyr with an area-weighted mean of 128 ± 34 mm/kyr and similar median value (122 mm/kyr). The Nankai watershed has the lowest rates of erosion, 21 – 83 mm/kyr, with an area-weighted mean of 50 ± 15 mm/kyr and median of 48 mm/kyr.

Considering the study area as a whole, erosion rates are significantly and positively correlated with mean basin slope, mean local relief, median $k_{sn}$, fraction of the watershed covered by forest, and fraction of the watershed covered by grassland or shrubland ($R^2 \geq 0.18$, p < 0.05, Figure 6). In the Yongchun watershed, erosion is significantly correlated with




slope, fraction of the watershed that is forest, and fraction of the watershed that is grass/shrubland ($R^2 \geq 0.32$, $p < 0.05$); correlations are positive with forest and slope and negative with grass/shrubland. In the Weiyuan watershed, erosion is significantly correlated with median $k_{sn}$, mean annual precipitation (MAP), fraction of the upstream watershed that is agriculture, fraction of the upstream watershed that is forested, mean basin slope, and mean local relief ($R^2 \geq 0.20$, $p < 0.05$);

relationship is positive for all terms except MAP and fraction of the upstream watershed that is forested. In the Nankai watershed, erosion is significantly correlated with MAP and fraction of the upstream basin that is grass/shrubland ($R^2 \geq 0.42$, $p < 0.01$); MAP is an inverse correlation while grass/shrubland is positive.

## 5 Discussion

Topographic and climatic parameters are often invoked as controls on long-term, [10]Be$_i$-determined erosion rates

(e.g., Portenga and Bierman, 2011) while human land use is assumed not to alter isotopically-determined erosion rates except in the most disturbed locations (Reusser et al., 2015). In small watersheds in Yunnan, where modern Chinese policies encouraged widespread deforestation and expansions in agriculture from the 1950s to 1980s, we find that topographic metrics correlate with erosion rates over the entire study area, with secondary effects that we interpret stem from human modification of the landscape. Mean annual precipitation is not important in setting erosion rates in our study area. Below we

explore the evidence for topographic control of erosion rates, the inverse correlations with rainfall, and evidence for human-induced increases in apparent [10]Be-determined erosion rates.

### 5.1 Topographic influence on erosion rates

Across the entire study area, we find that long-term erosion rates correlate best with measures of topographic steepness. Individual regressions for erosion rates as a function of topographic steepness (slope, relief, and median $k_{sn}$) are all

significant ($p < 0.01$) and combining all three parameters increases the $R^2$ to 0.62, suggesting that topographic steepness explains most of the variability in measured, long-term erosion rates in our study area (Figure 6).

Considering the basins individually, we see a decline in the influence of topographic parameters on erosion rates from north to south in the study area. The Yongchun River watershed (northernmost basin) has the strongest topographic signal in erosion rate patterns and long-term landscape evolution appears to be the primary control on erosion rates. Faulting

and base level fall from Mekong River incision created a transient landscape with streams draining three different sub-landscapes: high-elevation low-relief, actively-adjusting, and the footwall of the fault (Figure 7a). Mixing of sediment from the low-erosion rate, high-elevation, low-slope landscape with sediment from the more rapidly eroding, actively-adjusting and footwall landscapes has resulted in erosion rates that scale non-linearly with elevation, a proxy for proportion of low-slope area (Figure 7b). Basins with mean elevations >3000 m have lower long term erosion rates, 12 – 38 mm/kyr, than

basins draining proportionally less relict landscape (mean elevations <3000 m), 50 – 209 mm/kyr, confirmed by a Wilcoxon





rank-sum test (p = 0.003). Similar results have been found using $^{10}Be_i$ elsewhere in landscapes adjusting to baselevel fall (Willenbring et al., 2013), and in measured $^{10}Be_i$ erosion rates above and below knickpoints (Miller et al., 2013).

In contrast to other studies comparing $k_{sn}$ and erosion rate (Ouimet et al., 2009;Vanacker et al., 2015;DiBiase et al., 2010), we observe a non-linear decrease in erosion rates as mean basin $k_{sn}$ increases (Figure 7c). However, as found in other studies (e.g., Ouimet et al., 2009;Vanacker et al., 2015;Granger et al., 1996;Binnie et al., 2007;DiBiase et al., 2010;Montgomery and Brandon, 2002), including a global meta-analysis (Harel et al., 2016), erosion rates increase non-linearly with increasing mean basin slope (Figure 7c). This implies that erosion rates lag behind channel steepening and only increase after channel incision has lowered hillslope base-level and steepened slopes. Compared to a global compilation of erosion rates as a function of mean basin $k_{sn}$ (Harel et al., 2016), we find that the Yongchun has slightly lower erosion rates for the calculated $k_{sn}$ values. This could be a function of how $k_{sn}$ was calculated, as we used a higher resolution topographic dataset than Harel et al. (2016).

Topography also appears to exert a first order control on erosion rates in the Weiyuan watershed. We observe modest and statistically significant relationships ($R^2 = 0.20 – 0.36$, $p < 0.05$) between erosion rate and relief, mean slope, and median $k_{sn}$ (Figure 6). Combining relief and slope in a multiple regression increases $R^2$ to 0.52 ($p < 0.05$), but adding median $k_{sn}$ does not further improve the regression. Thus, although neither relief nor slope dominates the signal, it appears that topographic steepness terms explain a large percent of the variance observed in erosion rates in the Weiyuan watershed.

In contrast to the influence of topography in the Weiyuan and Yongchun watersheds, we see no influence of topography on long-term erosion rates in the Nankai watershed ($p > 0.05$ for regressions with all topographic parameters). One possibility is that the Nankai has had steady long-term base level, thus removing the influence of topography on erosion rates (e.g., Riebe et al., 2000). However, in this tectonically active area on the margins of the Tibetan Plateau, faults are mapped throughout all three watersheds (Burchfiel and Chen, 2012). Thus, this lack of correlation suggests to us that in the Nankai watershed erosion rate is controlled by other factors. We explore this supposition below.

**5.2 Inverse correlations with rainfall**

In contrast to other studies (e.g., Bierman and Caffee, 2001, 2002;Henck et al., 2011;von Blanckenburg, 2005), we find that mean annual precipitation does not correlate with erosion rates in our study area. There is no gradient in rainfall over the Yongchun watershed due to its small size relative to pixels of the APHRODITE dataset. There are significant correlations between erosion rate and mean annual precipitation for the both the Weiyuan and Nankai watersheds ($R^2 \geq 0.35$, $p < 0.01$), and for the two watersheds considered together ($R^2 = 0.72$, $p < 0.01$); these correlations are all inverse relationships in which erosion rates decrease as precipitation rates increase – an improbable result in terms of process considerations (Figure 6). We do not consider these inverse correlations to be a causal relationship but instead interpret them as a sign that rainfall covaries with slope, relief, and $k_{sn}$ ($R^2 = 0.69$, $p < 0.01$, and correlation is inverse for a multiple regression of mean annual precipitation as a function of relief and slope). Thus, we conclude that precipitation is not a significant control on erosion in this landscape.



### 5.3 Effects of agriculture on apparent $^{10}Be_i$ erosion rates

Our data suggest that in some small catchments, intensive land-use changes have increased apparent rates of erosion determined from $^{10}Be$ concentrations in fluvial sediment. If our explanation is correct, human influence increases from north to south in the study area, with no apparent influence of human activity on $^{10}Be_i$-determined erosion rates in the Yongchun watershed. In the Nankai watershed, the southernmost of our study area, we find that agricultural land use is the primary control on the pattern of erosion rates we measure. The Nankai watershed is heavily cultivated (48% of land use) and the original river channel is mostly obscured by these land-use changes; specifically, terraces that are flooded to grow rice and sugarcane cover the floor of the basin. Upland hillslopes in the Nankai watershed are impacted by land uses shown to result in elevated surficial erosion (Sidle et al., 2006), including forest conversion to agriculture, pasture, and tea and rubber plantations (Figure 2). Field observations suggest that much of the landscape mapped as grassland and shrubland is actually tea and rubber plantations. With this in mind, we interpret the positive correlation between the percent land cover of grass/shrubland and $^{10}Be_i$-determined erosion rate in the Nankai watershed as suggesting that the upland agriculture has increased erosion rates and is excavating sediment from below the mixed (turbated) soil layer and delivering it to channels (Figure 6).

By comparing sampled catchments in the Nankai with varying percentages of grass/shrubland but similar basin slope, we can make a first order estimate of the relative effect of agriculture on erosion rates (Table 1). For the watersheds where grass/shrubland increases from <0.05% to over 25%, erosion rates increased by an average factor of 2.5. However, for watersheds where less land was converted to grass/shrubland (0.06% to 9.3%), the upstream land use does not appear to affect erosion rates.

In the Weiyuan watershed, anthropogenic land use change appears to play a secondary role to topography in setting rates of erosion in the watershed. The regression of erosion rates as a function of slope and relief improves from $R^2 = 0.52$ to $R^2 = 0.62$ ($p < 0.01$) when fraction of the upstream watershed that is agricultural is added as an independent variable. This improved correlation suggests that in the Weiyuan watershed, agricultural land use plays a secondary, but still important, role in setting patterns of erosion rates. Comparing one watershed with only 6% agricultural land use to two watersheds with at least 20% agricultural land use but similarly steep slopes (~20°), agricultural land use increased apparent erosion rates by an average factor of 1.8, suggesting that steep landscapes are also susceptible to the effects of large-scale land use change.

The influence of human activity on long-term erosion rates calculated from $^{10}Be_i$ is unusual given that many other studies have found $^{10}Be_i$-derived erosion rates to be unaffected by anthropogenic activity unless the disturbance is deeper than ~30-60 cm (1995;Brown et al., 1998;Granger et al., 1996;Bierman and Steig, 1996;Hewawasam et al., 2003;Reusser et al., 2015;Vanacker et al., 2007). We thus conclude that in the Weiyuan and Nankai watersheds, agricultural activity must have resulted in the erosion of at least 30 cm of soil across large portions of each sampled watershed. In the Nankai watershed this excavation of material from depth has doubled apparent $^{10}Be$-determined erosion rates for the most heavily





altered watersheds. The watersheds we studied are relatively small (4-2508 km$^2$; mean = 309 km$^2$; median = 46.5 km$^2$), making it more likely that land use changes have altered large portions of the upstream areas. It is less likely that the effects of human activity would be detected in larger study areas where land use is less homogenous. We conclude that the most representative background rates of erosion will come from the least disturbed watersheds as well as larger watersheds where

heavily disturbed areas are less likely to cover significant areas.

## 6 Conclusions

Overall we find that erosion rates in Yunnan are correlated with both topographic and land use metrics. Our data and analysis suggests that human activity in the region has in some watersheds eroded sediment below the mixed layer and is sourcing deeply-derived sediment with lower concentrations of $^{10}Be_i$ to channels, thus increasing apparent erosion rates by as

much as a factor of 2.5. In heavily agricultural landscapes, such as the Nankai watershed, the effects of human activity have likely obscured the relationship between erosion rate and topographic parameters. In other locations, such as the Weiyuan watershed, both topography and human activity contribute to observed patterns of erosion. Thus, in small watersheds with extensive agriculture, erosion rates derived from *in situ* $^{10}Be$ can be affected by both topography and the intensity and distribution of human activity. This suggests that in other areas with high rates of human disturbance, *in situ* $^{10}Be$-derived

erosion rates could be inflated by human activity.

## Author contribution

AHS, TBN, PRB, and VSG planned the experimental design. AHS, TBN, and VSG conducted the field work. DHR completed AMS analyses. WBO completed $k_{sn}$ analyses. TBN completed all Be extractions. All authors contributed to analysing data and preparing the manuscript.

## Acknowledgments

This research was supported by NSF awards EAR-1114166 (to Schmidt), EAR-1114159 (to Bierman), and EAR-1114436 (to Rood). We thank the staff of the AMS Laboratory at the Scottish Universities Environmental Research Centre (SUERC) for support during $^{10}Be_i$ measurements. We thank C. M. Zhang, R. J. Wei, and J. A. Bower for field assistance. The data used in this paper are included as tables in the supplemental information.

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



**Table 1**

Comparison of Nankai samples with similar slopes but varying upstream grass/shrubland.

| Mean basin slope of comparison (°) | Low grass/shrub | | High grass/shrub | | Ratio of erosion rates[1] |
|---|---|---|---|---|---|
| | Percent grass/shrub | Erosion [mm/kyr] | Percent grass/shrub | Erosion [mm/kyr] | |
| 15.5-15.8 | 0.03 | 30 | 27.8-29.6 (n = 2) | 48-83 | 1.6-2.8 (mean = 2.18) |
| 16.1-16.4 | 0.01 | 21 | 30.3-41.6 (n = 3) | 40-65 | 1.9-3.1 (mean = 2.63) |
| 19.0-19.3 | 0.06 | 44 | 9.3 (n = 1) | 43 | 0.98 |

[1] Ratio calculated by dividing the apparent erosion rate for the landscape with high grass/shrub area by the erosion rate for the landscape with the low grass/shrub land use. The mean reported in parentheses is the mean for that slope range. The average of all 5 watersheds where grass/shrubland increased to over 25% is 2.5.



**Figure 1**

Study location. Inset shows the region of interest within Southeast Asia (a), and primary panels show the locations of the basins sampled, topography (b) (NASA LP-DAAC, 2012b), and mean annual precipitation (c) (Yatagai et al., 2012).

**Figure 2**

Field photographs that show representative land use and landscape in each of the watersheds studied. From north to south: the Yongchun watershed headwaters (A) and looking across the southern arm of the basin (D); the Weiyuan watershed agricultural land near valley bottoms (B) and cleared land along the mainstem of the Weiyuan river (E); the tea plantations and rice paddies in the headwater areas of the Nankai watershed (C) and the flat agricultural landscape and incised channel in the downstream parts of the watershed (F).

**Figure 3**

Elevation, hillslope angle, median basin $k_{sn}$ and land use for the study watersheds. Elevation, slope angle and $k_{sn}$ are derivatives of 30 m ASTER DEM's (NASA LP-DAAC, 2012b), and land use is provided by GLC30 (Chen et al., 2015). Mean basin $k_{sn}$ depicted for entire contributing area of each sample. Pie charts show the percentage of each land-use category in the basin, with the percentage of the most prevalent land-use noted on the chart. In the Yongchun watershed, the

yellow line denotes the boundary between adjusting and relict landscape, and the red line denotes an east-down normal fault (dotted where inferred from field observations, solid where mapped (Ministry of Geology and Mineral Resources, 1986)). Data shown in this figure are in table S4.

**Figure 4**

Map shows the location, basin boundary, and sample ID for each sample in the Yongchun (a), Weiyuan (b), and Nankai (c).

Data shown in this figure are in table S1.

**Figure 5**

Maps showing erosion rate for the entire contributing area at each sample site. Bar-and-whisker plots show the distribution of erosion rates and erosion indices by basin. The lower and upper limits of the central boxes indicate the 1[st] and 3[rd] quartiles, respectively, the central line indicates the median, and the whiskers extend to 1.5 times the inter-quartile range. Data shown

in this figure are in table S4.





**Figure 6**

Erosion rate as a function of (from left to right in columns) slope, mean annual precipitation (MAP), mean local relief, median $k_{sn}$, % of the landscape that is agricultural, % of the landscape that is forested, and % of the landscape that is grass and shrubland. From top to bottom, rows are all the data, the Yongchun data, the Weiyuan data, and the Nankai data. Numbers on the plots are $R^2$ values. * represents a significant correlation at $p < 0.05$, ** represents a significant correlation at $p < 0.01$. Grey plots are not significant ($p > 0.05$). Data shown in this figure are in table S4; regression statistics are in table S5.

**Figure 7**

Channel profiles for sampled sub-basins within the Yongchun watershed and erosion rates by distance from the outlet (A). Channel profiles end where the upstream area falls below ~0.54 km$^2$. Scatter plots show relationships between elevation, slope, and erosion rate (B), and median $k_{sn}$ and erosion rate (C). Data shown in this figure are in table S4.



Fig01

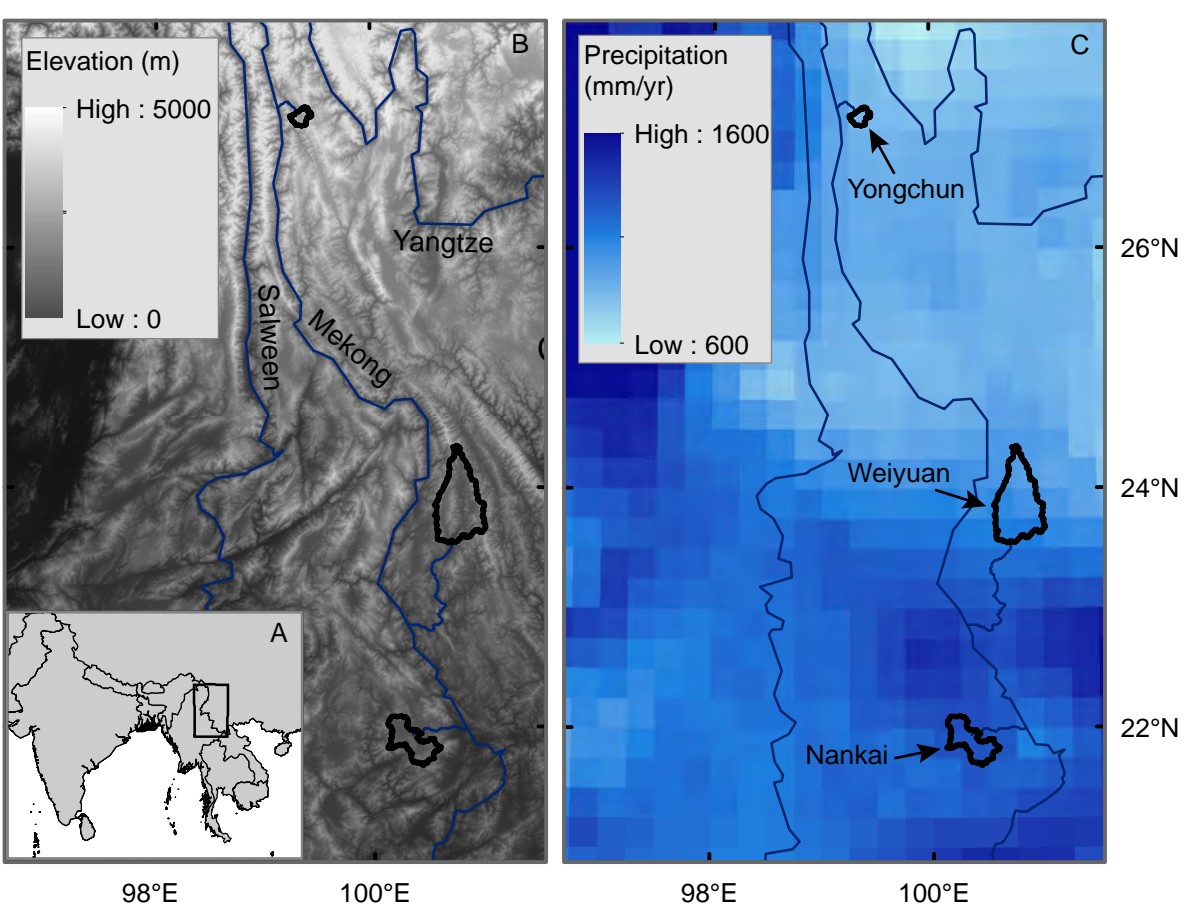



Fig02

| Yongchun | Weiyuan | Nankai |

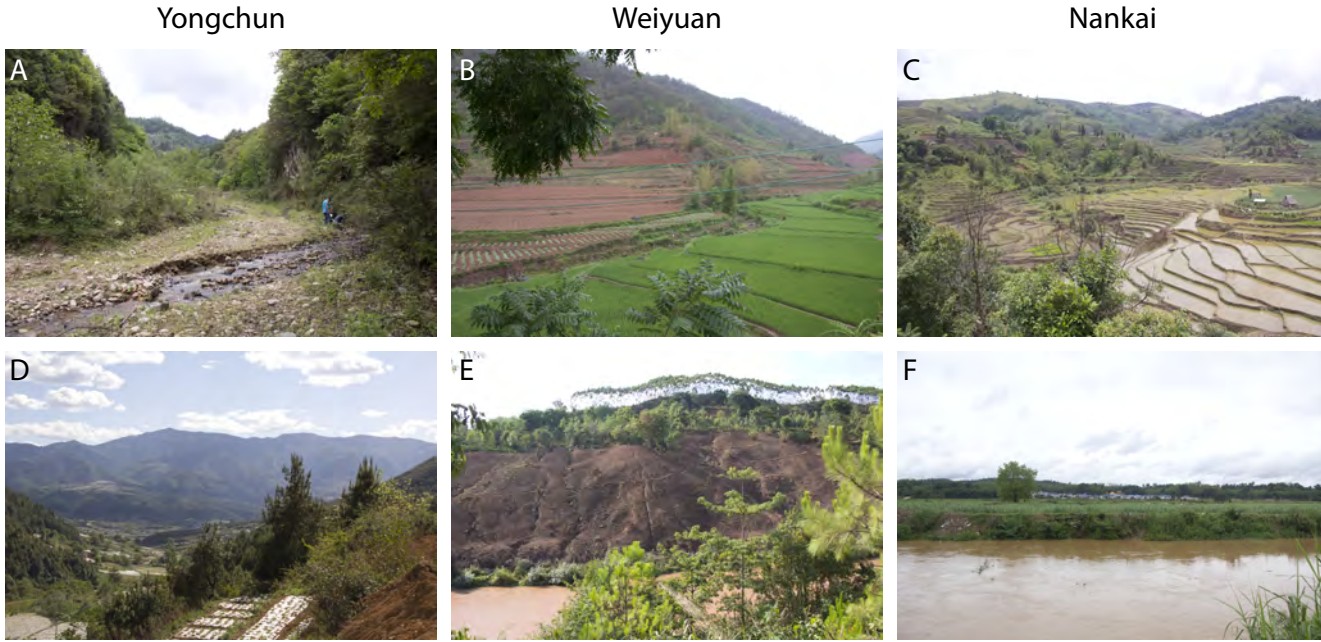





Fig03





Fig04




Fig05

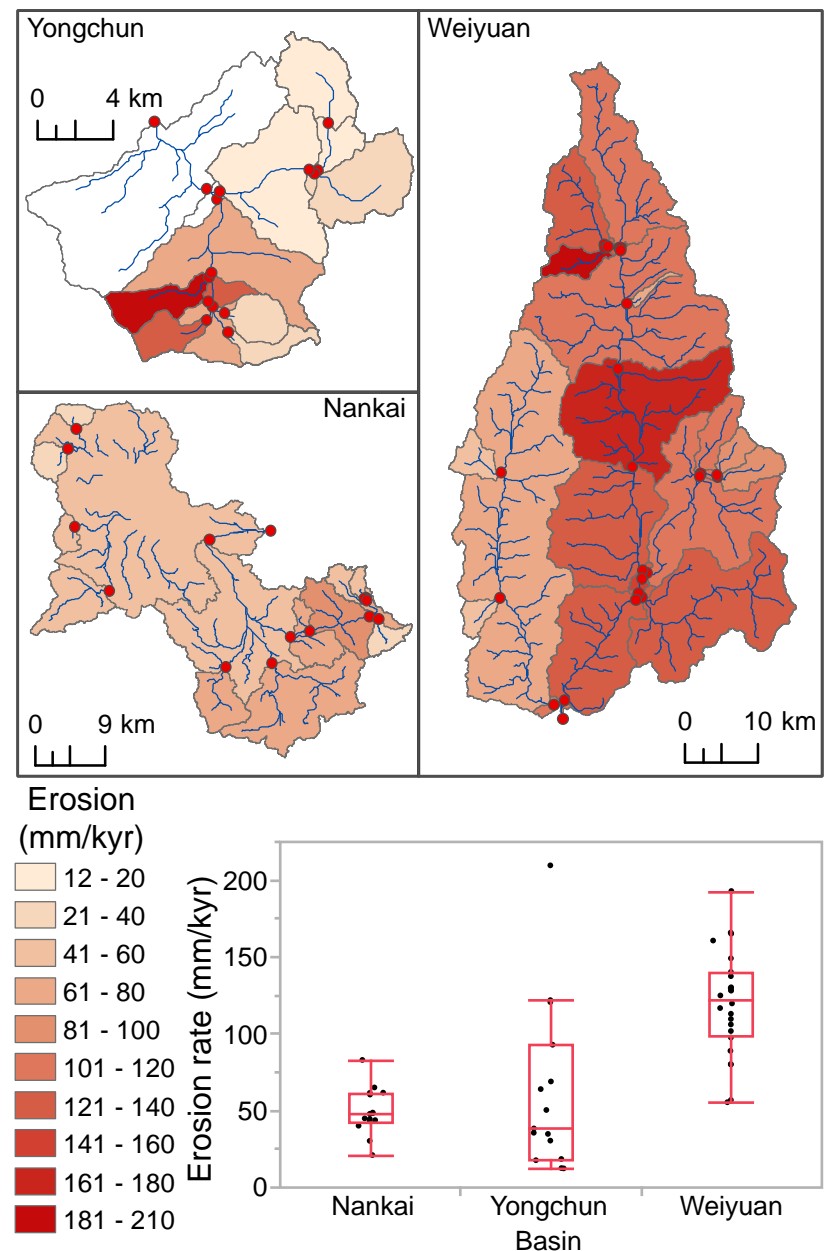

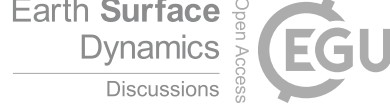



Fig06

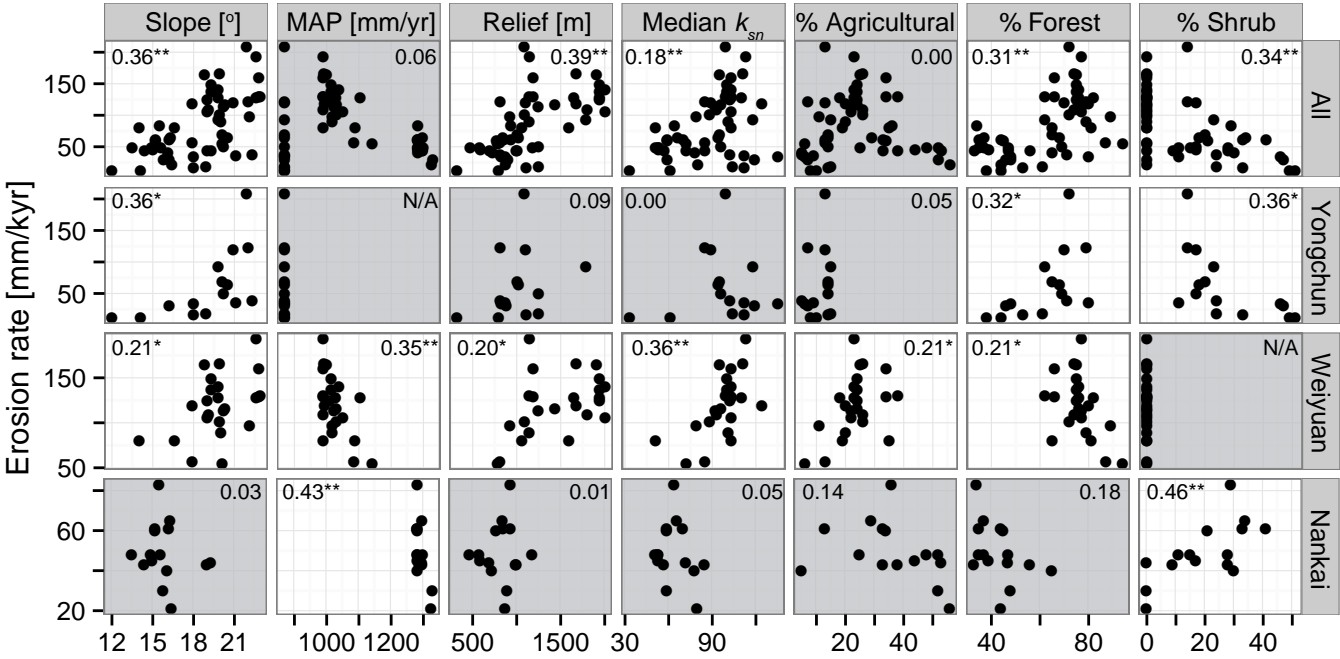



Fig07

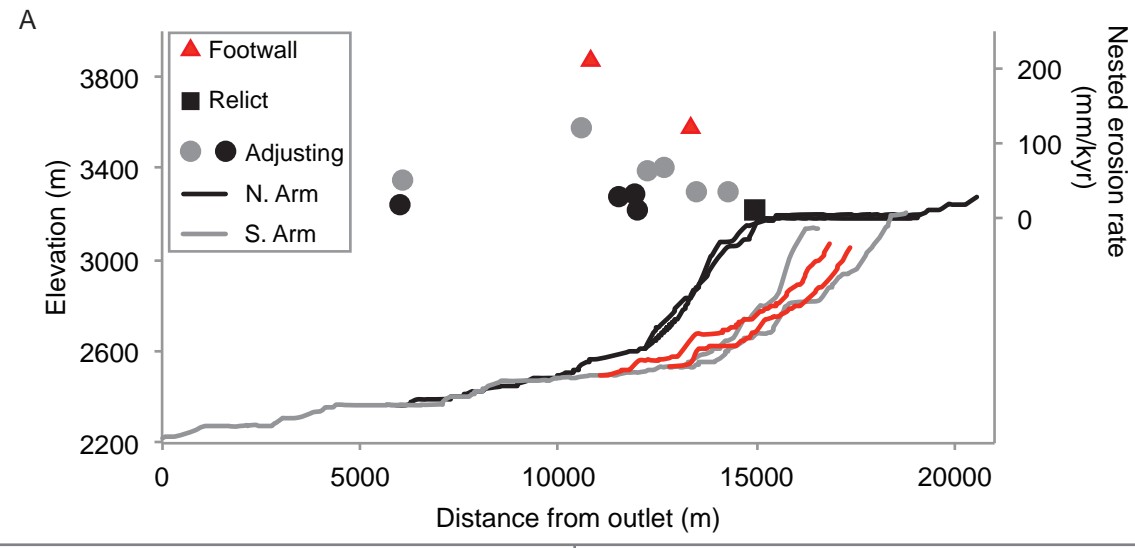

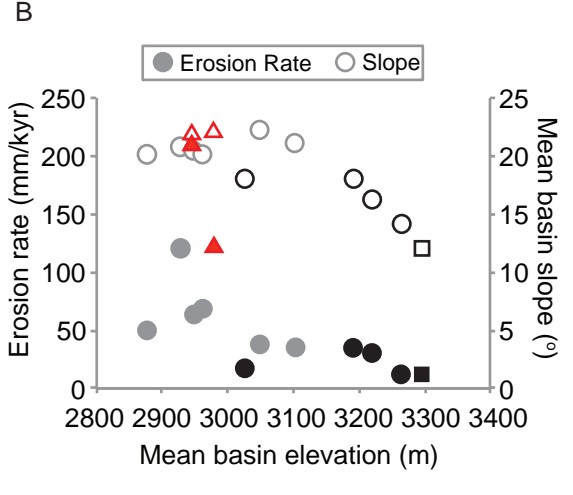

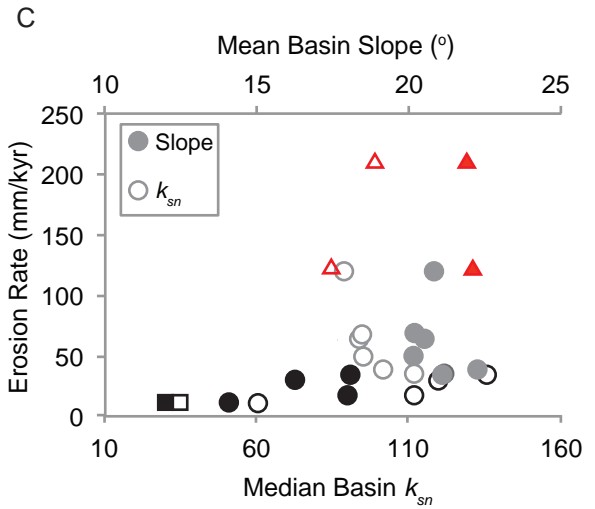