# Peer review of "Influence of topography and human activity on erosion in Yunnan, SW China"

_Earth Surface Dynamics, 2016_

## Referee Comment (RC1) · Anonymous Referee #1 · 22 Aug 2016

Schmidt et al. present an interesting study in which they aim to evaluate the effects of human activity on long-term apparent erosion rates calculated from measurements of in-situ 10Be collected from river sediments in a nested sampling scheme across three watersheds in Yunnan, China. The authors choose three watersheds differing in size across covarying gradients of elevation, precipitation, channel steepness and land use. The authors present the results of correlation and regression analyses of in-situ 10Be-derived long-term apparent erosion rates calculated using the CRONUS-Earth online calculator with topographic (basin average slope, normalized channel steepness and relief/elevation), climatic (mean annual precipitation), and land use/land cover variables.

Through this analysis, the authors present results which suggest that in-situ 10Be derived apparent erosion rates are related predominantly to topographic variables in two

of the three watersheds (Yongchun and Weiyuan) which are not dominated by human land-use), while apparent erosion rates are not correlated with topographic variables (and instead correlated with area in agricultural land use) for a third watershed (Nankai) which was heavily dominated by human land-use. Further regression analysis suggests that human land-use may play a secondary but not controlling role in the Weiyuan watershed (which lies in the middle of the land use gradient from most intensive human land-use (Nankai) to least intensive human land use (Yongchun).

The authors conclude that in-situ 10Be apparent erosion rates can be significantly affected by human land-use at the basin scale and suggest caution to researchers attempting to derive long term apparent erosion rates in watersheds which have undergone significant human land-use change, are not large enough to buffer the effects of land-use change on sediment composition, and are heavily disturbed.

This is a nicely packaged study that is worthy of publication. Although the results are not surprising (there are some watershed and basins in the midwestern and southeastern United States, for example, where a significant amount of the current surficial material is actually subsoil exposed due to the effect of highly erosive historical land use practices (Piedmont, Corn Belt Plains, Driftless Area)), they are important to consider for researchers who are considering constructing studies of long-term apparent erosion rates. However I do have a few suggestions which may assist the authors in improving the manuscript.

General comments: 1. In section 5.3 of the discussion, lines 11-15, the authors present anecdotal evidence from field observations that "much of the landscape mapped as grassland and shrubland is actually tea and rubber plantations...", and thus this land use category is actually reflective of agricultural land use. This is an absolutely critical observation which allows the reader to better interpret the data from the scatterplots presented in Figure 6, and it came to me as a surprise that it is buried in the discussion. I believe instead that this observation should be presented much earlier in the manuscript, perhaps in the methods section. Do the authors have any point data

or observations associated with this that they could include in the supplementary information? Have the authors attempted to sum fraction of agricultural land use and fraction of shrub/grassland (which would presumably represent "total" agricultural land use) and plot that against 10Be derived erosion rates? Is that relationship stronger or weaker than the relationships of erosion rates with agricultural land use and shrub alone? If stronger, it would suggest that the sum of agricultural land use (regardless of management practice) is important. If weaker it might suggest that the erosive effects of tea and rubber plantations on the uplands alone are extremely important in understanding the effects of these different land uses on apparent erosion rates. Additionally, although the relationship between %shrub and erosion rate in positive for Nankai, it seems to be negative for Yongchun. Does this imply that the land cover classified as shrub/grassland in Yongchun is truly shrub, whereas the land cover classified as shrub/grassland in Nankai is in fact tea and rubber plantations? Do the authors have any observations to assist in resolving this question? I believe any further interpretations that the authors can provide in this matter will greatly assist the readers in understanding the results presented, especially when attempting to link portions of the results/discussion to the Figures and Table. Table 1 should be presented with the results and moved out of the discussion.

2. I understand the authors' intent in Figure 7, however it took quite a bit of time to interpret. It may just be my own perspective, however I believe that interpretability may be improved by putting the erosion rate and slope data points into separate portions of the figure? Additionally, the text in P7, Lines 3-4 states that "...we observe a non-linear decrease in erosion rates as mean basin ksn increases..." (again, based on the figure, I believe the authors intend to write "median basin ksn"). However, in looking at Figure 7C, I would argue that (at least in my interpretation), erosion rates appear to show a peaked non-linear distribution as median basin ksn increases, not a non-linear decrease only.

3. The authors spend a significant amount of text discussing correlation numbers,

however I could not find text describing which correlation method was used (pearson, kendall, spearman, etc...). Also, it might be helpful (whether in the main manuscript or supplementary material) to include a table of correlations of the variables with eachother so that the reader could better put the patterns seen in the scatterplots of figure 6 in context.

4. In the manuscript text, the authors integrate regression statistics into the results and discussion to support some of their major conclusions. Some of these regressions (from the text) appear to have been multiple linear regressions, however in Table S5, the results that are displayed appear to be from simple linear regression. It would be helpful to include results and more details from multiple linear regressions in table S5. For example, were they conducted with or without interactions between variables?

Minor/Editorial comments: P1. Line 15 (Abstract): Perhaps should read; "In order to better understand..." instead of "In order to understand better..."? P1. Line 32: Missing names on first reference: "1995" P5. Line 15 (Eqn 1): I believe Eqn one should have a theta term included to represent concavity. P7. Line 4: The figure referenced in this sentence shows Median basin ksn, however the text reads mean basin ksn. P7. Line 32-33: Perhaps the authors might soften the assertion that "Thus, we conclude that precipitation is not a significant control on erosion in this landscape" with something like: "Thus, within the scale and scope of our study, we conclude that precipitation is not a significant control on erosion in this landscape". P8. Lines 16-25: These seem to belong in the results section and not the discussion section. P8. Line 30: Missing names on first reference: "1995". Figure 4. It appears that sample CH-01 and CH-23 which are included in supplementary table 1 are not included in Figure 4. A short explanation for why (or a footnote either in the supplementary tables or figure) would be helpful. Supplementary Tables: It would be helpful to include the basin name (instead of just the basin number) in the supplementary tables, as any reader who wishes to analyze or interpret the raw data will be familiar with the basin names only from the manuscript text.

**ESurfD**

Interactive
comment

---

## Referee Comment (RC2) · Anonymous Referee #2 · 3 Sep 2016

This manuscript is generally well-written and it was interesting to read this work which highlights the influence of topographical and climatic factors, and human intervention on catchment-wide erosion rates measured using in-situ produced cosmogenic 10Be. In the paper, authors have attempted to evaluate the control of climate, topographic steepness (mean basin slope, mean basin relief and normalized channel steepness) and land use changes that had occurred over a period of 30 years from 1950 to 1980 in the Yunnan River basin of SW China. They have selected three catchments within this large river basin, which is characterized by different geomorphological settings, a significant gradient in the precipitation pattern from the upper part (north) to lower part (south) and a noteworthy increase in human disturbances from the north to the south. Erosion rates in this river basin have been measured in fluvial sediments sampled from 52 small catchments. These samples represent a large array of topographic setting,

different rainfall regimes and end member as well as mixed situations of land use types. Locations for the sampling sites have been perfectly selected, which complies with the main scope of the research. Topographic parameters in the river basin have been derived digitally with the aid of a digital elevation model. Following a wide statistical analysis, correlations between the measured erosion rates and topographic factors, land use types and mean annual precipitation have been deduced.

Even though the rainfall significantly varies from north to south of the river basin, a control of mean annual precipitation on erosion rates has not been detected in this landscape. Overall, they have found a correlation between the erosion rates and topographic factors and land use types for the entire river basin. Correlation between the topographic steepness and erosion rate is strong for the Yongchun catchment, which is situated in the north, and then it has decreased from the north-most catchment (minimally disturbed by the agriculture) to the south-most catchment (significantly disturbed by agriculture). They have found that in the south-most catchment (Nankai), there is no significant influence of topography on cosmogenic nuclide derived erosion rates suggesting that erosion in this catchment is mainly controlled by human perturbation since it is the highest disturbed catchment for agriculture. Further, they claim that cosmogenic nuclide derived erosion rates in small catchments within the Nankai catchment have increased up to 2.5 as a result of agricultural land use. Subsequently, the main conclusion of this research points to small catchments in the river that are under intensified agriculture erode sediments below the mixed layer of the landscape by exporting sediments to the fluvial system with low cosmogenic nuclide concentrations. However, they have not adequately discussed on how this mixed layer is developed within the landscape and then become homogenized in in-situ cosmogenic nuclides. This is the main theoretical outline of their finding, which should be highlighted. Nevertheless, they have not cited the relevant papers in the literature that discuss the same phenomenon in many landscapes elsewhere in the world. Even though their correlation analyses support this argument of sending sediments below the mixed layer, which is not connected to the field observations. Therefore, more explanations on catchment characteristics and especially hillslope erosional mechanisms and how these sediments are derived below the mixed layer need to be presented to strengthen their argument.

I would like to give following three main suggestions to further enhance the quality of their manuscript.

1. Include additional information on In-situ produced cosmogenic nuclide method / Formation of mixed layer / Possibility of eroding sediments below the mixed layer in agricultural catchments referring to the literature.

Cosmogenic nuclide derived erosion rates have been used as background erosion rates even in the perturbed catchments because sediments are usually derived below the mixed layer in many landscapes worldwide. Presence of mixed layer in a few landscapes is documented in previous studies, thickness of this layer is considered as 50-100 cm in the tropics but can be up to 3–4 m (van Breemen and Buurman, 1998; Wielemaker, 1984). At steady state, the cosmogenic nuclide concentration of the mixed layer becomes equal to that of its surface (Brown et al., 1995; Granger et al., 1996). In situ produced cosmogenic homogeneity in this mixed layer was experimentally illustrated in many landscapes (Braucher et al., 2000, Small et al., 1999; Schaller et al., 2002b). Deriving of sediments below the mixed layer in perturbed catchments, and if so, possibility of estimating higher apparent erosion rates have also been discussed in previous studies (von Blanckenburg et al., 2004). Therefore, authors are advised to revisit the relevant literature since this paper is biased to the fact that sediments are eroded below the mixed layer in highly agricultural catchments.

2. Provide more catchment characteristics/ Erosional mechanisms.

Since the main aim of this manuscript is to show that cosmogenic-nuclide derived erosion rates in some small catchments are affected by recent changes in land use, it is important to present more field evidences to support this hypothesis. For example, it would be interesting if you are able to show that sediments in these disturbed catchments are now derived below the mixed layer either by sheet erosion or they are eroded

below the mixed layer via linear erosion (rills, gullies or landsliding). In the latter case, an appropriate portion of the catchment should be eroding under linear processes to increase the catchment-wide erosion rates in the perturbed catchment by a factor 2.5. This phenomenon has been revealed by Von Blanckenburf et al., 2004 using a simple model. It would be important for authors to elaborate the mechanisms of how the sediments with low cosmogenic nuclide concentrations are derived below the mixed layers in the agricultural catchments.

3. Provide details of Quaternary alluvial deposits.

Under geology, authors have simply mentioned that Yongchun and Nankai catchments are largely covered by Quaternary alluvial sediments. But, it has not been mentioned whether sediments are also derived or not from these Quaternary formations. If the catchments contain large quantities of recent sediments, there is a possibility of re-working these alluvial sediments from the storages to the river in addition to delivering sediments from hillslopes. If the sediment storage within the catchments is large and occurs for a longer period it does significantly affect the net cosmogenic concentration that you have measured in fluvial sediments. This is because sediments sampled from rivers (point bars, mid channel islands, river bed, etc.) should have accumulated additional cosmogenic nuclides by storing within the catchment. In contrast, there is a possibility for sediments to lose their concentration by radioactive decay during burial in alluvial deposits depending on their age. Therefore, if your sediments were not contaminated with sediments from alluvial storages, it is important to mention. This justification will build up the forcefulness of your subsequent analyses to evaluate how erosion rates are affected by topography, climate and land use.

Minor Comments P1, line 24 – What is the depth of tillage? It should be very deep if you argue that sediments are derived below the mixed layer by tillage. P 2, Line 2 – I don't think that the readers who are not much familiar in cosmogenic nuclide method will really understand what is meant by mixed layer. P2 Line 4 – this is not an assumption. Cosmogenic nuclide homogenization has been experimentally demonstrated in many

landscapes elsewhere. P2, Lines 31-32 Need more details about Quaternary alluvium P2, lines 33 and P3, line 24-25– Why the authors are concerned to know the relative distribution of quartz in the watershed. I don't think that geological map will provide this information. P 6, lines 1-2 – Not clear P6, line 29, Authors have used the term "long-term erosion rates" through the manuscript. I think this term is relative, hence may not be appropriate to use since they don't have any short-term erosion rate presented in the manuscript. For those who work on much longer time scales, cosmogenic nuclide-derived erosion rates seem to be short-term. Therefore, the term "cosmogenic nuclide-derived erosion rate" may be more appropriate than 'long-term erosion rate". P 8, line 13-15, This looks like an assumption. This fact can be justified using data/filed observation. P8, line 30, I think that these depth of the mixed layer should be garter than 30-60 cm. P9, line 4, For larger watershed, there are some other issues when the cosmogenic method is applied. Temporary storage of sediments in larger watershed has to be considered. P 13, Figure caption – (a), (b) and (c) should be capitalized P 21, Figure 07 has not been sufficiently discussed in the text

---

## Author Comment (AC1) · 19 Sep 2016

We received two careful and thorough reviews from anonymous reviewers. Below are their comments (in black) and our response to their comments (in blue). We have addressed all concerns raised by the reviewers.

**Anonymous Referee #1**

Schmidt et al. present an interesting study in which they aim to evaluate the effects of human activity on long-term apparent erosion rates calculated from measurements of in-situ 10Be collected from river sediments in a nested sampling scheme across three watersheds in Yunnan, China. The authors choose three watersheds differing in size across covarying gradients of elevation, precipitation, channel steepness and land use. The authors present the results of correlation and regression analyses of in-situ 10Be-derived long-term apparent erosion rates calculated using the CRONUS-Earth online calculator with topographic (basin average slope, normalized channel steepness and relief/elevation), climatic (mean annual precipitation), and land use/land cover variables.

Through this analysis, the authors present results which suggest that in-situ 10Be derived apparent erosion rates are related predominantly to topographic variables in two of the three watersheds (Yongchun and Weiyuan) which are not dominated by human land-use), while apparent erosion rates are not correlated with topographic variables (and instead correlated with area in agricultural land use) for a third watershed (Nankai) which was heavily dominated by human land-use. Further regression analysis suggests that human land-use may play a secondary but not controlling role in the Weiyuan watershed (which lies in the middle of the land use gradient from most intensive human land-use (Nankai) to least intensive human land use (Yongchun).

The authors conclude that in-situ 10Be apparent erosion rates can be significantly affected by human land-use at the basin scale and suggest caution to researchers attempting to derive long term apparent erosion rates in watersheds which have undergone significant human land-use change, are not large enough to buffer the effects of land-use change on sediment composition, and are heavily disturbed.

This is a nicely packaged study that is worthy of publication. Although the results are not surprising (there are some watershed and basins in the midwestern and south- eastern United States, for example, where a significant amount of the current surficial material is actually subsoil exposed due to the effect of highly erosive historical land use practices (Piedmont, Corn Belt Plains, Driftless Area)), they are important to consider for researchers who are considering constructing studies of long-term apparent erosion rates. However I do have a few suggestions which may assist the authors in improving the manuscript.

General comments: 1. In section 5.3 of the discussion, lines 11-15, the authors present anecdotal evidence from field observations that "much of the landscape mapped as grassland and shrubland is actually tea and rubber plantations...", and thus this land use category is actually reflective of agricultural land use. This is an absolutely critical observation which allows the reader to better interpret the data from the scatterplots presented in Figure 6, and it came to me as a surprise that it is buried in the discussion. I believe instead that this observation should be presented

much earlier in the manuscript, perhaps in the methods section. Do the authors have any point data or observations associated with this that they could include in the supplementary information? Have the authors attempted to sum fraction of agricultural land use and fraction of shrub/grassland (which would presumably represent "total" agricultural land use) and plot that against 10Be derived erosion rates? Is that relationship stronger or weaker than the relationships of erosion rates with agricultural land use and shrub alone? If stronger, it would suggest that the sum of agricultural land use (regardless of management practice) is important. If weaker it might suggest that the erosive effects of tea and rubber plantations on the uplands alone are extremely important in understanding the effects of these different land uses on apparent erosion rates. Additionally, although the relationship between %shrub and erosion rate in positive for Nankai, it seems to be negative for Yongchun. Does this imply that the land cover classified as shrub/grassland in Yongchun is truly shrub, whereas the land cover classified as shrub/grassland in Nankai is in fact tea and rubber plantations? Do the authors have any observations to assist in resolving this question? I believe any further interpretations that the authors can provide in this matter will greatly assist the readers in understanding the results presented, especially when attempting to link portions of the results/discussion to the Figures and Table. Table 1 should be presented with the results and moved out of the discussion.

The observation about tea and rubber plantations is made in the Field Sites section (section 2). We have expanded this a bit and made it more obvious. This is done in lines 13-14 of page 4. This is based on field observations, which is now clarified. The other watersheds do not have the same "problems" with the land use mapping – field observations suggest that forests are mostly forest, shrub/grassland is mostly shrub/grassland and so on. Fraction agricultural land does not have a significant correlation with erosion in the Nankai watershed ($R^2 = 0.14$, p = 0.17), possibly because the agricultural land is primarily on very flat areas where we have few samples due to difficulty in locating the river channel. This also could be explained by the reviewer's hypothesis – the upland agriculture (the shrub/grassland) is the big eroder, not the downstream terraced agriculture. We have added this more nuanced perspective to the discussion in lines 26-29 of page 8.

When we add the fraction agricultural land and the fraction shrub/grassland together for Nankai, the correlation is not significant ($R^2 = 0.17$, p = 0.13). Similarly, although a multiple regression of erosion as a function of agricultural and grass/shrubland increases increases slightly ($R^2 = 0.55$) compares to a single regression against just grass/shrubland, the p value for the independent agriculture parameter is not significant (p = 0.50). Adding forest to the regression, for a multivariable linear regression with three independent parameters increases the regression very little ($R^2 = 0.56$) and with three independent parameters and only 15 sample sites, the p values for all variables are not significant (p > 0.63). In addition, there is a complication that the three variables co-vary. Urban area is only a tiny fraction of the study site and otherwise, land is classified as forested, grass/shrub, or agricultural. Thus, when we start to combine the three primary land use categories of the watersheds, we lose the differences between watersheds.

2. I understand the authors' intent in Figure 7, however it took quite a bit of time to interpret. It may just be my own perspective, however I believe that interpretability may be improved by

putting the erosion rate and slope data points into separate portions of the figure? Additionally, the text in P7, Lines 3-4 states that "...we observe a non- linear decrease in erosion rates as mean basin ksn increases..." (again, based on the figure, I believe the authors intend to write "median basin ksn"). However, in looking at Figure 7C, I would argue that (at least in my interpretation), erosion rates appear to show a peaked non-linear distribution as median basin ksn increases, not a non-linear decrease only.

This is an excellent observation. We have revised the figure and fixed the explanation on line 15 of page 7.

3. The authors spend a significant amount of text discussing correlation numbers, however I could not find text describing which correlation method was used (pearson, kendall, spearman, etc...). Also, it might be helpful (whether in the main manuscript or supplementary material) to include a table of correlations of the variables with each other so that the reader could better put the patterns seen in the scatterplots of figure 6 in context.

We have added this explanation to the methods (digital data, section 3) on lines 28-29 of page 5. We also added table S6 to show correlations among other variables.

4. In the manuscript text, the authors integrate regression statistics into the results and discussion to support some of their major conclusions. Some of these regressions (from the text) appear to have been multiple linear regressions, however in Table S5, the results that are displayed appear to be from simple linear regression. It would be helpful to include results and more details from multiple linear regressions in table S5. For example, were they conducted with or without interactions between variables?

The expanded explanation of regressions on lines 28-29 of page 5 addresses multiple regressions as well. In addition, we have added the results of the multiple regressions discussed in the text to table S5.

Minor/Editorial comments:

P1. Line 15 (Abstract): Perhaps should read; "In order to better understand..." instead of "In order to understand better..."?

We have not changed this because the most common standards of English writing suggest that you should not split infinitives. This is somewhat disputed, but it seems safer to go with avoiding split infinitives.

P1. Line 32: Missing names on first reference: "1995"

This has been fixed.

P5. Line 15 (Eqn 1): I believe Eqn one should have a theta term included to represent concavity.

Yes, good catch. It was lost converting the manuscript to the Copernicus template and I have returned it.

P7. Line 4: The figure referenced in this sentence shows Median basin ksn, however the text reads mean basin ksn.

Yes, good catch. We've fixed this.

P7. Line 32-33: Perhaps the authors might soften the assertion that "Thus, we conclude that precipitation is not a significant control on erosion in this landscape" with something like: "Thus, within the scale and scope of our study, we conclude that precipitation is not a significant control on erosion in this landscape".

We agree – this study is quite small. This has been fixed.

P8. Lines 16-25: These seem to belong in the results section and not the discussion section.

Although we see how this could go in the results section, We feel more comfortable leaving it in the discussion section. The discussion is about the effects of agriculture while the results simply report the various correlations.We think there is some flexibility on what can go in each section and this provides context for the discussion but would be disjointed from the results section.

P8. Line 30: Missing names on first reference: "1995".

Fixed.

Figure 4. It appears that sample CH-01 and CH-23 which are included in supplementary table 1 are not included in Figure 4. A short explanation for why (or a footnote either in the supplementary tables or figure) would be helpful.

We have removed these samples. They are downstream of a big dam and were removed from the analysis because of the effects of the dam. It was a mistake to have them in table 1.

Supplementary Tables: It would be helpful to include the basin name (instead of just the basin number) in the supplementary tables, as any reader who wishes to analyze or interpret the raw data will be familiar with the basin names only from the manuscript text.

Thanks for catching this oversight. We have put names in all the tables.

**Anonymous Referee #2**

This manuscript is generally well-written and it was interesting to read this work which highlights the influence of topographical and climatic factors, and human intervention on catchment-wide erosion rates measured using in-situ produced cosmogenic 10Be. In the paper, authors have

attempted to evaluate the control of climate, topographic steepness (mean basin slope, mean basin relief and normalized channel steepness) and land use changes that had occurred over a period of 30 years from 1950 to 1980 in the Yunnan River basin of SW China. They have selected three catchments within this large river basin, which is characterized by different geomorphological settings, a significant gradient in the precipitation pattern from the upper part (north) to lower part (south) and a noteworthy increase in human disturbances from the north to the south. Erosion rates in this river basin have been measured in fluvial sediments sampled from 52 small catchments. These samples represent a large array of topographic setting, different rainfall regimes and end member as well as mixed situations of land use types. Locations for the sampling sites have been perfectly selected, which complies with the main scope of the research. Topographic parameters in the river basin have been derived digitally with the aid of a digital elevation model. Following a wide statistical analysis, correlations between the measured erosion rates and topographic factors, land use types and mean annual precipitation have been deduced.

Even though the rainfall significantly varies from north to south of the river basin, a control of mean annual precipitation on erosion rates has not been detected in this landscape. Overall, they have found a correlation between the erosion rates and topo- graphic factors and land use types for the entire river basin. Correlation between the topographic steepness and erosion rate is strong for the Yongchun catchment, which is situated in the north, and then it has decreased from the north-most catchment (minimally disturbed by the agriculture) to the south-most catchment (significantly disturbed by agriculture). They have found that in the south-most catchment (Nankai), there is no significant influence of topography on cosmogenic nuclide derived erosion rates suggesting that erosion in this catchment is mainly controlled by human perturbation since it is the highest disturbed catchment for agriculture. Further, they claim that cosmogenic nuclide derived erosion rates in small catchments within the Nankai catchment have increased up to 2.5 as a result of agricultural land use. Subsequently, the main conclusion of this research points to small catchments in the river that are under intensified agriculture erode sediments below the mixed layer of the landscape by exporting sediments to the fluvial system with low cosmogenic nuclide concentrations. However, they have not adequately discussed on how this mixed layer is developed within the landscape and then become homogenized in in-situ cosmogenic nuclides. This is the main theoretical outline of their finding, which should be highlighted. Nevertheless, they have not cited the relevant papers in the literature that discuss the same phenomenon in many landscapes elsewhere in the world. Even though their correlation analyses support this argument of sending sediments below the mixed layer, which is not connected to the field observations. Therefore, more explanations on catchment characteristics and especially hillslope erosional mechanisms and how these sediments are derived below the mixed layer need to be presented to strengthen their argument.

I would like to give following three main suggestions to further enhance the quality of their manuscript.

1. Include additional information on In-situ produced cosmogenic nuclide method / Formation of mixed layer / Possibility of eroding sediments below the mixed layer in agricultural catchments referring to the literature.

Cosmogenic nuclide derived erosion rates have been used as background erosion rates even in the perturbed catchments because sediments are usually derived be- low the mixed layer in many landscapes worldwide. Presence of mixed layer in a few landscapes is documented in previous studies, thickness of this layer is considered as 50-100 cm in the tropics but can be up to 3–4 m (van Breemen and Buurman, 1998; Wielemaker, 1984). At steady state, the cosmogenic nuclide concentration of the mixed layer becomes equal to that of its surface (Brown et al., 1995; Granger et al., 1996). In situ produced cosmogenic homogeneity in this mixed layer was experimentally illustrated in many landscapes (Braucher et al., 2000, Small et al., 1999; Schaller et al., 2002b). Deriving of sediments below the mixed layer in perturbed catchments, and if so, possibility of estimating higher apparent erosion rates have also been discussed in previous studies (von Blanckenburg et al., 2004). Therefore, authors are advised to revisit the relevant literature since this paper is biased to the fact that sediments are eroded below the mixed layer in highly agricultural catchments.

This is a very good point. We have added a paragraph with a more thorough introduction to the mixed layer to page 2 (lines 3-8).

2. Provide more catchment characteristics/ Erosional mechanisms.

Since the main aim of this manuscript is to show that cosmogenic-nuclide derived erosion rates in some small catchments are affected by recent changes in land use, it is important to present more field evidences to support this hypothesis. For example, it would be interesting if you are able to show that sediments in these disturbed catchments are now derived below the mixed layer either by sheet erosion or they are eroded below the mixed layer via linear erosion (rills, gullies or landsliding). In the latter case, an appropriate portion of the catchment should be eroding under linear processes to increase the catchment-wide erosion rates in the perturbed catchment by a factor 2.5. This phenomenon has been revealed by Von Blanckenburg et al., 2004 using a simple model. It would be important for authors to elaborate the mechanisms of how the sediments with low cosmogenic nuclide concentrations are derived below the mixed layers in the agricultural catchments.

We don't think we really have the data or field observations to say for certain what is driving the deep erosion. I suspect that it is simply a long history of agricultural land use without appropriate erosion control measures. For example, we frequently saw agricultural land all the way to rivers with furrows running parallel to the hillslope, rather than contouring it. Clearly if this type of land use is pervasive in a watershed, it can cause the kind of sheetwash that would result in sediment from below the mixed layer being delivered to the river. We have added some of this speculation to the discussion and also cited the appropriate and important paper you mention. This is in a paragraph on pages 8-9 (from line 30 on page 8 to line 7 on page 9).

3. Provide details of Quaternary alluvial deposits.

Under geology, authors have simply mentioned that Yongchun and Nankai catchments are largely covered by Quaternary alluvial sediments. But, it has not been mentioned whether sediments are also derived or not from these Quaternary formations. If the catchments contain large quantities

of recent sediments, there is a possibility of reworking these alluvial sediments from the storages to the river in addition to delivering sediments from hillslopes. If the sediment storage within the catchments is large and occurs for a longer period it does significantly affect the net cosmogenic concentration that you have measured in fluvial sediments. This is because sediments sampled from rivers (point bars, mid channel islands, river bed, etc.) should have accumulated additional cosmogenic nuclides by storing within the catchment. In contrast, there is a possibility for sediments to lose their concentration by radioactive decay during burial in alluvial deposits depending on their age. Therefore, if your sediments were not contaminated with sediments from alluvial storages, it is important to mention. This justification will build up the forcefulness of your subsequent analyses to evaluate how erosion rates are affected by topography, climate and land use.

In the Nankai watershed, only two samples (CH-071 and CH-075) are taken from areas with extensive Quaternary sediments. The other samples are from upland locations. This is because in the lowland areas, rice paddies and irrigation were so common as to nearly entirely obscure the natural river channel in the Nankai watershed.

In the Yongchun watershed, most of the Quaternary fill is downstream of our farthest downstream sample, with only sample CH-024 possibly sourcing some of the fill sediments. We cannot estimate how much of this sample has sediments from the fill material.

We have added two sentences to the field sites section (section 2) explaining what I outline above. They are lines 7 on page 2 and 10-11 on page 3.

Minor Comments
P1, line 24 – What is the depth of tillage? It should be very deep if you argue that sediments are derived below the mixed layer by tillage.

In the abstract we say that tillage and resultant erosion are sourcing deep sediments. We don't know how deep tilling is and it seems likely that part of the increase in erosion is due to sheet wash in areas with reduced ground cover due to erosion and tilling.

P 2, Line 2 – I don't think that the readers who are not much familiar in cosmogenic nuclide method will really understand what is meant by mixed layer.

This has been clarified with an additional paragraph as explained in response to your point 1, above.

P2 Line 4 – this is not an assumption. Cosmogenic nuclide homogenization has been experimentally demonstrated in many landscapes elsewhere.

This has been clarified with an additional paragraph as explained in response to your point 1, above.

P2, Lines 31-32 Need more details about Quaternary alluvium

This has been clarified with an additional paragraph as explained in response to your point 3, above.

P2, lines 33 and P3, line 24-25– Why the authors are concerned to know the relative distribution of quartz in the watershed. I don't think that geological map will provide this information.

We are concerned about the relative abundance of quartz because we measure $^{10}$Be in quartz. If the landscape is not shedding quartz in proportion to its erosion rate, for example, if there is widespread carbonate in the watersheds, then erosion rate estimates are not going to be accurate. We are explaining in these lines that we cannot quantify quartz distribution in the watershed.

P 6, lines 1-2 – Not clear

This is now p 6, lines 11-12. This sentence, which starts on line 10, is presenting correlation results for the Yongchun watershed. We have tried to clarify it.

P6, line 29, Authors have used the term "long- term erosion rates" through the manuscript. I think this term is relative, hence may not be appropriate to use since they don't have any short-term erosion rate presented in the manuscript. For those who work on much longer time scales, cosmogenic nuclide-derived erosion rates seem to be short-term. Therefore, the term "cosmogenic nuclide-derived erosion rate" may be more appropriate than 'long-term erosion rate".

This has been fixed throughout the manuscript.

P 8, line 13-15, This looks like an assumption.  This fact can be justified using data/field observation.

This is now page 8 lines 23-25. We do not have the soil profile data or detailed maps of erosion patterns to test this hypothesis. Thus, we suggest that the reason for the correlation is due to elevated erosion in response to agriculture, but we cannot formally test that with field data such as suggested by von Blanckenburg (2004).

P8, line 30, I think that these depth of the mixed layer should be garter than 30-60 cm.

Again, we have no data on the actual depth of the mixed layer at this location. We could guess that it is thicker because the location is sub-tropical, but without soil profile data to complement the basin average erosion rates, we cannot make more than general statements about how much material has been lost.

P9, line 4, For larger watershed, there are some other issues when the cosmogenic method is applied. Temporary storage of sediments in larger watershed has to be considered.

This is definitely true and we have added a caveat to be concerned with sediment storage.

P13, Figure caption – (a), (b) and (c) should be capitalized

This has been fixed.

P21, Figure 07 has not been sufficiently discussed in the text

In response to concerns from reviewer 1, figure 7 has been clarified. This also resulted in an extended explanation in the text about the figure (p 7, lines 3-22).

[revised manuscript text omitted]

Fig02

| Yongchun | Weiyuan | Nankai |
| --- | --- | --- |

[Figure]

Fig03

[Figure]

Elevation above sea level | Hillslope angle | Normalized channel steepness | Land-use

Yongchun

0 5 km

62%

Weiyuan

0 20 km

77%

Nankai

0 20 km

48%

Elevation (m)
Low : 700    High : 4000

Slope (°)
Low : 0    High : 30

Median $k_{sn}$
<60  60-80  80-100  100-120  >120

Cultivated    Forest    Grassland
Wetland    Water    Artifical
Shrubland

Fig04

[Figure]

Fig05

[Figure]

Fig06

[Figure]

[Figure]